# The Associations of Meteorological and Environmental Factors with Memory Function of the Older Age in Urban Areas

**DOI:** 10.3390/ijerph19095484

**Published:** 2022-04-30

**Authors:** Yuehong Qiu, Zeming Deng, Chujuan Jiang, Kaigong Wei, Lijun Zhu, Jieting Zhang, Can Jiao

**Affiliations:** 1School of Psychology, Shenzhen University, Shenzhen 518060, China; 1950482005@email.szu.edu.cn (Y.Q.); zemingdeng@126.com (Z.D.); 2070481006@email.szu.edu.cn (K.W.); zhulijunjxnu@163.com (L.Z.); 2Center for Mental Health, Shenzhen University, Shenzhen 518060, China; 3School of Music and Dance, Division of Arts, Shenzhen University, Shenzhen 518060, China; yatou_1218@163.com

**Keywords:** older age, memory function, meteorological and environmental factors, mixed effects model

## Abstract

Individual, meteorological, and environmental factors are associated with cognitive function in older age. However, little is known about how meteorological and environmental factors interact with individual factors in affecting cognitive function in older adults. In the current study, we used mixed effects models to assess the association of individual, meteorological, and environmental factors with cognitive function among older adults in urban areas. Data from 2623 adults aged 60 to 91 years from 25 provinces (or autonomous regions/municipalities) from the China Family Panel Studies (CFPS) were used. We used the memory test in CFPS to measure memory function, while meteorological data from the daily climate data set of China’s surface international exchange stations, and the traffic and greening data compiled by the National Bureau of Statistics (NBS) of China, were used to assess meteorological and environmental factors. The ICC of the empty model indicated that 7.7% of the variation in memory test scores for the older adults was caused by provincial characteristics. Results showed that the temperature and relative humidity of provinces moderated the effect of gender on the memory function for the older urban adults. Specifically, in the high temperature areas, memory scores for females were higher than those of males, and in the middle humidity areas, memory scores were also higher for the females than those of males. This study explained how meteorological and environmental factors played roles in influencing demographic factors on memory function among older adults. Further research is needed to better define the role and potential mechanism of this moderation.

## 1. Introduction

According to the United Nations (UN), in the 2018 Revision of World Urbanization Prospects report, the global urban population will increase by 2.5 billion by 2050, by which time nearly 70% of the population will live in cities [1].

At the same time, with the acceleration of the global population aging process, the financial and social burdens which are brought by cognitive impairment (e.g., mild cognitive impairment and Alzheimer’s disease) have become increasingly severe [2]. Memory decline, the earliest and most important symptom of cognitive impairment in the elderly, has become an important factor affecting the quality of later life [3,4]. Therefore, it is necessary to explore the factors that influence the memory function of older urban adults.

Many factors affect the memory function of older adults. A large amount of evidence has shown that individual risk factors are associated with memory decline in older adults [5,6,7]. The memory function of the elderly will decline with age [8], while chronic disease (e.g., diabetes and cardiovascular disease) is an additional risk factor for memory impairment in the elderly [9,10]. Older adult females are more likely to suffer from memory impairment than older adult males [11,12]. In addition, the education level and occupation [13], family economic conditions [14], and nutritional level [15] are all related to the decline of memory function in the elderly.

A large number of studies have confirmed that the impact of living in urban areas on human memory function is related to meteorological factors (e.g., temperature, precipitation, illumination, and other natural phenomena related to atmospheric motion) [16,17,18]. For example, some studies have shown that cold or hot temperatures influence memory functions in older adults [19,20,21], while others have not [22]. Although it is more likely that extreme temperature is associated with poor cognition, the elderly usually live in an environment with stable meteorological factors, rather than in extreme environments. Among these factors, temperature and humidity are thought to be the most stable and sensitive [23,24]. Therefore, temperature and humidity have often been highlighted and used as independent predictors of cognitive performance [16,19,25,26]. Other meteorological indicators, such as ground temperature and atmospheric pressure, are essentially affected by temperature and humidity, so they have serious multicollinearity problems [27]. Some studies have pointed out that this association may originate from the influence of meteorological factors on the blood oxygen level of human brain, and these factors may subsequently inhibit the activity of brain regions including the frontal lobe, occipital lobe, temporal lobe, and limbic system [28,29,30].

In addition, there is a close association with environmental factors (e.g., greening and traffic) and memory decline in older adults [31,32,33]. This can be explained by the fact that greening or traffic (e.g., highway construction) can directly or indirectly influence the surrounding microclimate and ecological environment [34,35]. So far, the evidence on the relationship between these factors and age-related memory decline is inconsistent. Only considering single meteorological factors and/or ignoring environmental factors is one of the reasons for these inconsistent results [20,36]. Additionally, small sample sizes (*n* < 30) can also affect the significance of the results [22]. The limitations of statistical methods, such as not accurately estimating the associations between meteorological factors (i.e., macro factors) and individual factors (micro factors), would also lead to the deviation of the observed results.

To fill this knowledge gap, the objective of the current study was to investigate whether meteorological and environmental factors were associated with memory function in older urban adults by an analytical strategy suitable for examining multi-level influences (i.e., micro and macro). In this study, the monthly average temperature, the relative humidity, and the greening space of each province in China were taken as the variables at the provincial level. Age, gender, and chronic disease (e.g., diabetes and hypertension) of the older adults were taken as individual variables. Additionally, the traffic level (highway construction) was taken as the covariate. Cognitive function of the older adults was reflected by the memory test scores. It has been pointed out in the literature that meteorological and environmental factors have different effects on demographic factors. For example, the same range of temperature change has different effects on people of different genders or ages [31,37,38]. Therefore, it is necessary to explore the moderation between meteorological/environmental factors and individual factors.

We used a mixed model, as shown in Figure 1, to test the hypotheses that the variables at the cross-level interactions had main and moderating effects on memory function in older adults. The mixed model can not only accurately analyze large-sample nested data, but also improve the estimation of effects at various levels. It estimates parameters more effectively than traditional regression analysis and is most suitable for testing cross-level predictors.

We hypothesized that individual variables (i.e., age, gender, and chronic disease) would be associated with memory function in older adults (H1). The meteorological and environmental variables would be associated with memory function in older adults (H2). The meteorological and/or environmental variables would moderate the relationships between individual variables and memory function in older adults (H3).

## 2. Methods

### 2.1. Data Sources and Participants

In this study, the data we used were extracted from the China Family Panel Studies (CFPS), a national large-scale screening data set with provinces (or autonomous regions/municipalities) as the unit. The cognitive tests were conducted on different participants in 2012 and 2016. Hence, memory tests in 2012 and 2016 were included in the current study.

Geographic location and interview date were used to accurately match the memory test scores with the local meteorological and environmental data at the month of the test. The meteorological data in this study was from the daily climate data set of China’s surface international exchange stations (Surface_Climate Data_China_ Multiple Elements_Daily Value Data_Climate Exchange Station, SCCMDC, V3.0). The data set contained daily records of meteorological conditions from 166 monitoring stations in China. The greening data, and the highway data in traffic, came from the China Statistical Yearbook of 2013 and 2017, which was compiled by the National Bureau of Statistics (NBS) of China (http://www.stats.gov.cn, accessed on 18 June 2021).

We included data for participants aged 60 years or above, who provided us with demographic information including gender, age, residence, years of education (excluding illiteracy), and chronic diseases (excluding mild cognitive impairments and different forms of dementia), and who completed the memory test score. We excluded participants who did not have information on the interview date and on their province of residence.

In total, 2623 older adults from 25 provinces (including autonomous regions/municipalities) participated in the study. The demographic information of all the participants was shown in Table 1.

### 2.2. Variable Measurement 

#### 2.2.1. Individual Level Variables

The variables of the individual level mainly included age, gender, and chronic disease. The information was extracted from the demographic information recorded in CFPS. In this study, gender and chronic disease were used as categorical variables, and age was used as a continuous variable.

#### 2.2.2. Provincial Level Variables

At the provincial level, we mainly focused on the following two meteorological variables: monthly average temperature and relative humidity in 2012 and 2016. The statistical regulations were as follows: (1) the daily average temperature and relative humidity were the average values of four time observations (2:00, 8:00, 14:00, 20:00); (2) When a certain fixed time value was not measured, the daily average value of corresponding elements was not measured. The hourly observation data from the ground automatic station. uploaded in real time. had passed the station quality control. The National Meteorological Information Center of China input the observation data and station quality control codes, uploaded in real time, into the real-time database for the data service (http://data.cma.cn, accessed on 15 February 2022). The main environmental variable we investigated was the green space of cities, which was obtained from the NBS.

We matched provincial level meteorological and environmental data with CFPS samples in the following way. First, we extracted the month and the location (on the provincial level) that the participants were interviewed from the CFPS. Second, the monthly average values of temperature and humidity of each province in the corresponding month were extracted from SCCMDC, and the greening data of the corresponding year were extracted from NBS. Finally, meteorological and environmental data were matched with the memory scores of subjects in CFPS according to the dates and provinces.

#### 2.2.3. Outcome Variable

The memory test in CFPS was used to measure the memory function of the older adults. The prototype of the test is from the Health and Retirement Study (HRS).

In the memory test, the interviewer read out 10 common words (mountains, rice, rivers, etc.) to the participant. After hearing all 10 words, the participant was asked to immediately recall the words read by the interviewer. The score of this recall was called the instant memory score. After a few minutes, the interviewer would ask the participant to recall the 10 words which just heard again. The score of this recall was called the delayed memory score. The memory scores were calculated by summing the total number of words correctly answered by the older adults, without considering the order. More details were introduced in the prior published paper [39] and on the CFPS website (http://www.isss.pku.edu.cn/cfps, accessed on 18 June 2021).

#### 2.2.4. Covariate

It is known that various air pollutants have specific and mixed impacts on human cognition. However, these pollutants were not reported comprehensively for all provinces in the CFPS data-set. According to the previous literature, for the elderly living in urban areas, motor vehicles are major emitters of PM2.5 and nanoparticles [40], while the density of arterial traffic is closely related to the degree of air pollution in residential areas [41]. Therefore, traffic can be used as a surrogate marker for various forms of air pollution and has been recognized as such [34,35,40,41].

Highway construction at a provincial level was used as covariate. It refers to the total length of public transit (such as bus lanes, etc.) and rail transit in a province. We used the China Statistical Yearbook of 2013 and 2017, for which the data was compiled by the NBS of China, to calculate the average highway mileage of 25 provinces. We matched traffic data according to the matching principles of the meteorological and environmental data and the CFPS individual’s data.

### 2.3. Data Analysis

In this study, a mixed effects model was used to evaluate the effects of different level variables on memory performance in older adults. In the 2012 and 2016 CFPS raw data, a total of 72,611 volunteers participated in this large-scale survey. According to our screening criteria (see Section 2.1 Data sources and Participants), this study included 2623 older adults from 25 provinces in China. Statistical assumptions associated with MEM (i.e., normality of the residuals, linearity, multicollinearity, and undue influence) were checked and were met for all models (see Appendix A).

The memory test score of CFPS was used as the outcome variable (Y_ij_). According to the previous studies [27,42], the model was established via the following steps:

Firstly, an empty model (i.e., a null model) was established to test whether there was significant difference in memory scores of the older adults in different provinces. According to Cohen (1988) [43], the mixed model is appropriate when the intraclass correlation coefficient (ICC) is greater than 0.059 [44]. In addition to the size of ICC, the design effect is a function of the intraclass correlation and the average cluster size [45]. It serves the following two purposes: (a) estimation of sample size, and (b) appraising the efficiency of complex designs. For cluster sampling samples, the design effect is usually greater than 1. In the mixed model, the individuals in each layer are homogeneous, and cluster sampling will cause a decrease in estimation efficiency. Therefore, the smaller the design effect value, the higher the accuracy [46].
The design effect = 1 + (average cluster size − 1) × intraclass correlation

The total variance of the outcome variable was divided into two parts, as follows: the between-province variance (*υ*_0*j*_), and the within-province variance (*ε*_*ij*_). Equation (1) indicated that the memory scores of each older adult individual (*Y_ij_*) could be estimated by the average memory scores of the older adults in all provinces (*γ*_00_), between-province variance (*υ*_0*j*_) and within-province variance (*ε_ij_*).
(1)$$\;\;\;\;\;\;\;\;\;\;\;\;\;\;\;\;\;\;\;\;Y_{ij}=\gamma_{00}+\upsilon_{0j}+\varepsilon_{ij}$$

Secondly, we established an individual level model (i.e., a level 1 model) to test the influence of individual level variables on memory scores. As shown in Equation (2), we added three variables at the individual level. The memory scores of each older adult individual (*Y_ij_*) could be estimated by the average memory scores of the older adults in all provinces (*γ*_00_), age (*γ*_10_), gender (*γ*_20_), and chronic disease (*γ*_30_), between-province variance (*υ*_0*j*_) and within-province variance (*ε_ij_*).
(2)$$Y_{ij}=\gamma_{00}+\gamma_{10}age_{j}+\gamma_{20}gender_{j}+\gamma_{30}chronic\;disease_{j}+\upsilon_{0j}+\varepsilon_{ij}\;$$

Thirdly, we established a model including both individual level and provincial level (i.e., a level 1 & 2 model) to test the effects of individual level variables and provincial level variables on memory scores of the older adults, concurrently. Because of the correlation among the meteorological factors (e.g., between air temperature and atmospheric pressure), considering them as predictors simultaneously will produce multicollinearity problems when analyzing their effects on memory function [47]. In the previous studies, temperature or relative humidity have been recognized as the main meteorological factors on cognitive function [48,49]. Thus, the provincial level variables included temperature (*γ*_01_), relative humidity (*γ*_02_), and greening space (*γ*_03_). In addition, a control variable was added, traffic condition (*γ*_04_). Therefore, as shown in Equation (3), the memory score of each older adult individual (*Y_ij_*) could be estimated by the average memory scores of the older adults in all provinces (*γ*_00_) and the effects of individual level variables (*γ*_10_, *γ*_20_ and *γ*_30_), provincial level variables (*γ*_01_, *γ*_02_, *γ*_03_ and *γ*_04_), and variances (*υ*_0*j*_ and *ε*_*ij*_).
(3)$$\begin{array}{l} Y_{\mathrm{ij}}=\gamma_{00}+\gamma_{10}{\mathrm{age}}_{j}+\gamma_{20}{\mathrm{gender}}_{j}+\gamma_{30}{\mathrm{chronic}\;\mathrm{disease}}_{j}\\ +\gamma_{01}{\mathrm{temperature}}_{j}+\gamma_{02}{\mathrm{humidity}}_{j}+\gamma_{03}{\mathrm{greening}}_{j}+\gamma_{04}{\mathrm{traffic}}_{j}\\ +\upsilon_{0j}+\varepsilon_{\mathrm{ij}} \end{array}$$

Finally, we established a full model, including main and moderating effects across levels. We tested the interaction between meteorological and environmental factors and demographic factors. As shown in Equation (4), (*γ*_11_) represented the moderating effect of the provincial level variable (temperature) on the relationship between an individual level variable (age), and the outcome variable (memory test score). Similarly, (*γ*_12_, *γ*_13_, *γ*_14_, *γ*_21_, *γ*_22_, *γ*_23_, *γ*_24_, *γ*_31_, *γ*_32_, *γ*_33_ and *γ*_34_) represented the moderating effects of provincial level variables on the relationship between individual level variables and the outcome variable.
(4)$$\begin{array}{ll} Y_{ij}=\gamma_{00}+\gamma_{10}age_{j} & +\gamma_{20}gender_{j}+\gamma_{30}chronic\;disease_{j}+\gamma_{01}temperature_{j}+\gamma_{02}humidity_{j}\\ & +\gamma_{03}greening_{j}+\gamma_{04}traffic_{j}+\gamma_{11}age_{j}\times temperature_{j}\\ & +\gamma_{12}age_{j}\times humidity_{j}+\gamma_{13}age_{j}\times greening_{j}+\gamma_{14}age_{j}\times traffic_{j}\\ & +\gamma_{21}gender_{j}\times temperature_{j}+\gamma_{22}gender_{j}\times humidity_{j}\\ & +\gamma_{23}gender_{j}\times greening_{j}+\gamma_{24}gender_{j}\times traffic_{j}\\ & +\gamma_{31}chronic\;disease_{j}\times temperature_{j}\\ & +\gamma_{32}chronic\;disease_{j}\times humidity_{j}+\gamma_{33}chronic\;disease_{j}\times greening_{j}\\ & +\gamma_{34}chronic\;disease_{j}\times traffic_{j}+\upsilon_{0j}+\varepsilon_{ij} \end{array}$$

We took age, temperature, humidity, greening, and highway construction as continuous variables. We set gender as a categorical variable, with male (marked as 1) and female (marked as 2). Years of education ≤ 9 means having received a basic education, excluding illiteracy, and >9 means reaching the level of senior high school or above. Chronic diseases (including diabetes, hypertension, and chronic cardiovascular disease) are divided into suffering (marked as 1) and not (marked as 0). The raw data was extracted by Stata (College Station, TX, USA) SE (v15.0), and IBM (Amonk, New York, NY, USA) SPSS statistics (v19.0) was used for data analysis.

## 3. Results

### 3.1. The Empty Model

The results in Table 2 showed that there were significant differences in memory scores of the older adults from different provinces. We calculated ICC considering both within-province variance and between-province variance. The ICC of the empty model was 0.077, which indicated that 7.7% of the memory test scores variation of the old age was caused by provincial characteristics (ICC = between-province variance/[between-province variance + within-province variance] = 0.755/[0.755 + 9.086]). In addition, the ICC result also showed that the data structure of this study was nested, so it was suitable for the mixed model.

### 3.2. The Individual Level Model

Results showed that the individual model significantly improved the fit to the data compared to the empty model (Table 2). The statistical results showed that the within-province variance of the individual level model (8.652) was smaller than that of the empty model (9.086). Moreover, the gap value (13,159.545) was also smaller than that of the empty model (13,280.283). The ICC of the individual model was 0.071, and the design effect value (8.38) was lower than that of the empty model (9.00).

The individual level model verified H1 that the memory test scores were related to the age and gender of the older adults. Specifically, in the elderly, older age (*γ*_10_ = −0.107, *t* = −11.242, *p* < 0.001) and male (*γ*_20_ = 0.233, *t* = 1.980, *p* = 0.048) were associated with lower memory scores (Table 3 and Table 4). The empty versus individual level models were assessed on the basis of the Restricted Maximum Likelihood (REML) test.

### 3.3. The Individual Level and Provincial Level Model

In this model, we examined the effects of individual and provincial level variables on the memory test scores of older adults. In the aspect of model fitting, the gap value of this model (13,185.948) was smaller than that of the empty model (13,280.283), but slightly larger than that of the individual level model (13,159.545). The within-province variance (8.623) was smaller than that of empty model (8.652) and individual level model (9.086). Table 2 showed that the ICC of this model was 0.067, and the design effect value (7.96) was lower than that of the empty model (9.00) and the individual level model (8.38).

The results confirmed the H1 again, as the results in Table 4 showed that older age (*γ*_10_ = −0.107, *t* = −11.268, *p* < 0.001) and male sex (*γ*_20_ = 0.237, *t* = 2.026, *p* = 0.043) were associated with lower memory scores, after controlling for the effects of the provincial level variables. Moreover, among the variables at the provincial level, humidity (*γ*_02_ = −0.049, *t* = −2.943, *p* = 0.003) was associated with the memory scores of older adults. Higher humidity was associated with lower memory test scores. These results partially confirmed the H2. Therefore, a full model should be analyzed, including the direct effects of two levels and the moderating effects of different levels to test the third hypothesis.

### 3.4. Interactive Effects between Individual and Provincial Variables on Memory Performance

As shown in Table 5, the average temperature moderated the effect of gender on the memory scores (*γ*_21_ = 0.049, *t* = 2.661, *p* = 0.008). According to the distribution of meteorological data in this study, we further divided the temperature into the following three levels: low-level temperature with less than 20 °C (250 cases), middle-level temperature with 20–25 °C (843 cases), and high-level temperature with more than 25 °C (1530 cases). Simple effect analysis showed that in high-level temperature, the memory scores of females were higher than that of males (*p* = 0.001), while in the middle and low-level, the memory scores were not significantly different between the two genders (shown in Figure 2a).

Besides, the average relative humidity moderated the effect of gender on the memory scores (*γ*_22_ = −0.049, *t* = −2.107, *p* = 0.035). We further divided the humidity into three levels: low-level humidity with less than 70% (458 cases), middle-level humidity with 70–79.9% (1609 cases), and high-level humidity with more than 80% (556 cases). Simple effect analysis showed that in the middle humidity, the memory scores were higher for the females than that of males (*p* = 0.002); while in the high and low humidity, the memory scores were not significantly different between the two genders (shown in Figure 2b).

These results partially confirmed the H3; meteorological factors of provinces had moderating effects on gender in the memory scores of older adults in urban areas. The results of the REML test for the full model were shown in Table 2.

## 4. Discussion

In this study, we used mixed effects models to assess the association of individuals and meteorological and environmental factors with memory function of older urban adults. Results showed that temperature and relative humidity of provinces moderated the effect of gender on the memory function for the older urban adults. Specifically, in the high temperature areas, the memory scores of females were higher than those of males. In the middle humidity, the memory scores were higher for the females than for males.

In the individual level and provincial level model, consistent with previous studies, there were significant gender differences in memory scores, with females outperforming males on auditory memory tasks [50]. Additionally, our study found that aging had a negative impact on the memory function of the elderly. This may be related to the age-related changes of gray matter and white matter in the brains of the elderly [51,52]. Furthermore, we found a negative correlation between humidity and memory scores of the elderly. Some studies have found that relatively high humidity environments will reduce the cognitive ability of individuals [25], making it more difficult for them to think clearly and reducing their alertness [26]. At the same time, the main effects of gender and age were still significant and, therefore, it is necessary to further examine how meteorological/environmental factors have moderating effects on individual variables in memory function.

Importantly, in the full model, we found the average temperature of provinces moderated the effect of gender on the memory function for the older urban adults. In the high-level temperature areas, the memory scores of females were higher than that of males, while in the low temperature areas, memory scores were not significantly different between the two genders. This is consistent with previous studies [16], which may be due to the fact that female generally prefer higher indoor temperatures than males [53,54].

One possible explanation is that mitochondrial dysfunctions manifest during normal aging [30]. Mitochondrial Deoxyribonucleic Acid (DNA) impacts brain function during aging through synaptic transmission [55]. Mitochondrial pathways directly regulate adaptive thermogenesis, which is defined as the production of heat in response to changes in environmental temperature, and which gradually decreases in elderly [30,56]. The adaptation of human body to heat stress is highly dependent on adaptive thermogenesis. Heat stress disrupts brain functional connectivity [57,58] and subsequently affects the moderating effect of mitochondrial DNA on the relationship between temperature and cognitive function [30]. Therefore, heat stress increases neuronal requirements for cognitive tasks in older adults [59,60]. The regulatory capacity of the elderly (such as cardiovascular function) gradually declines and, hence, they are more vulnerable to the interference of heat stress [48,61]. Compared with the older male, the older female has stronger adaptability to heat stress. Some studies report that this is due to the different physiological structures between males and females (e.g., lower body weight and surface area, and higher surface area-to-mass ratio). Thus, older female adults have better memory performance as compared to their male counterparts in high-level temperature areas.

Furthermore, our results showed that the relative humidity of provinces moderated the effect of gender on the memory function for the older urban adults. In the middle humidity, memory scores were higher for the females than for males, while in the high and low humidity, memory scores were not significantly different between the two genders. This may be related to the fact that males are more sensitive to humidity.

It has been proved that living in high humidity areas is related to the decline of cognitive function [25]. In this environment, the relative power of the θ-band, α-band, and β-band of people are significantly decreased. People feel sleepier, which make it more difficult for them to think clearly [26]. Some studies have pointed out that males sweat more than females in the same environment and, hence, it is reasonable to assume that the physiological and psychological states of males are more susceptible to humidity [38,62], which in turn influence their cognitive function [37,38]. At present, the research on gender difference and humidity needs to be verified by follow-up studies. We did not determine the role of traffic in the relationship between meteorological factors and cognitive function, which could not deny the influence of other pollutants on cognition. Moreover, the influence of traffic pollutants on cognition may be stronger in environments with high humidity [63,64].

The advantage of this study is the use of three large data sets with wide geographical coverage, avoiding the problem of an insufficient sample size [22]. Additionally, unlike previous studies which only considered single meteorological or environmental factors [20,36], we explored the association of meteorological and environmental factors with the memory function of older adults. Additionally, we used a mixed model to examine the micro and macro multi-level effects, which improved the estimation of effects at the individual level and provincial level, and estimated parameters more effectively than traditional regression analysis.

One of the limitations was that some individual factors (e.g., diet structure, physical activity. and mental state) and meteorological/environmental factors (e.g., noise and different kinds of air pollutants) could not be considered in this study due to not being available in the data-set and, thus, casual conclusion could not be made based on the current findings. Another limitation was that we used the data of 2012 and 2016 in CFPS as cross-sectional data, and did not consider the variation (e.g., economic and social development) during the period. Furthermore, because the meteorological data used in this study was accurate to the month, and the environmental data was accurate to the province, we could further investigate the comprehensive effects of climate, environment, and ecology, and report more precise dates and addresses. In future research, we will look for larger data sets or platforms with more information, and further investigate the comprehensive effects of climate, environment, and ecology, and report more precise dates and addresses. In addition, we will try to use different data analysis methods to explore the possibility of non-linear associations between environmental and climatic factors and cognition.

## 5. Conclusions

In this study, 2623 older urban adults in CFPS were selected to assess the association of individual, meteorological, and environmental factors with memory function by using mixed effects models. The main findings showed that temperature and relative humidity of provinces moderated the effect of gender on the memory function for the older urban adults. This study explained how the meteorological and environmental factors played roles in the influence of demographic factors on the memory function of the older adults, and it provided a reference for future research into cognitive aging.

## Figures and Tables

**Figure 1 ijerph-19-05484-f001:**
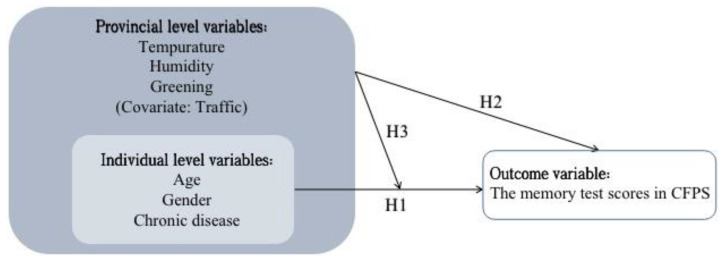
The proposed multilevel model.

**Figure 2 ijerph-19-05484-f002:**
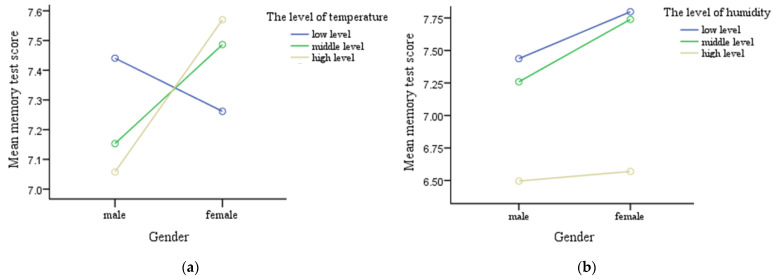
(**a**) Memory scores of different genders under different temperature levels. (**b**) Memory scores of different genders under different humidity levels.

**Table 1 ijerph-19-05484-t001:** Descriptive statistics of continuous and categorical variables.

Variables	All Population	
Continuous Variables	n	Mean ± SD (Min–Max)
Age	2623	67.5 ± 6.1 (60–91)
Memory test score	2623	7.29 ± 3.07 (0–14)
Temperature (°C)		24.60 ± 7.03 (−21.2–30.1)
Humidity (%)		74.97 ± 5.08 (52.2–88.8)
Greening (hm2)		39.04 ± 2.98 (30–48.4)
Traffic (km)		28,745.55 ± 23,516.00 (4907–102,707)
Categorical variables		
Gender	*N*	%
Male	1522	58.0
Female	1101	42.0
Education level		
Low (≤9)	1919	73.2
High (>9)	704	26.8
Cardiovascular disease		
Yes	860	32.8
No	1763	67.2

**Table 2 ijerph-19-05484-t002:** The gap value, variance, and ICC value of models.

	Gap (−2)	Variance		ICC	Design Effect
		Within-Province Variance	Between-Province Variance		
Null model	13,280.283	9.086	0.755	0.077	9.00
Level 1 model	13,159.545	8.652	0.665	0.071	8.38
Level 1 & 2 model	13,185.948	8.623	0.621	0.067	7.96
Full model	13,240.987	8.622	0.620	0.067	7.96

**Table 3 ijerph-19-05484-t003:** Descriptive statistics of memory scores of categorical variables.

Categorical Variables.	*n*	Mean ± SD
Gender		
male	1522	7.12 ± 3.039
female	1101	7.51 ± 3.101
Chronic disease		
Yes	860	7.19 ± 3.089
No	1763	7.33 ± 3.061
Total score	2623	7.29 ± 3.070

**Table 4 ijerph-19-05484-t004:** The main effects of the empty, individual level, and provincial level model.

	The Empty Model				The Individual Level Model				The Individual Level and Provincial Level Model			
Parameter	Estimate	*t*	*p*	95% CI	Estimate	*T*	*p*	95% CI	Estimate	*t*	*p*	95% CI
Intercept	7.008	36.348	<0.001	6.604–7.413	13.930	19.975	<0.001	12.561–15.297	19.935	10.111	<0.001	16.035–23.835
Individual level variable												
Age (*γ*_10_)					−0.107	−11.242	<.001	−0.126–−0.089	−0.107	−11.268	<0.001	−0.126–−0.089
Gender (*γ*_20_)					0.233	1.980	0.048	0.002–0.463	0.237	2.026	0.043	0.008–0.467
Chronic disease (*γ*_30_)					−0.066	−0.537	0.591	−0.309–0.176	−0.055	−0.441	0.660	−0.297–0.118
Provincial level variable												
Temperature (*γ*_01_)									0.009	0.861	0.390	−0.012–0.031
Humidty (*γ*_02_)									−0.049	−2.943	0.003	−0.082–−0.016
Greening (*γ*_03_)									−0.063	−1.524	0.134	−0.147–0.020
Control variable												
Traffic (*γ*_04_)									−3.75 × 10^−7^	−0.052	0.959	−1.53 × 10^−5^–1.45 × 10^−5^

**Table 5 ijerph-19-05484-t005:** The effects of individual-level and province-level variable in the full model.

	The Full Model					
Parameter	Estimate	SE	*df*	*t*	*p*	95% CI
Intercept	15.810	11.863	2571.133	1.333	0.183	−7.452–39.073
Individual-level variable						
Age (*γ*_10_)	−0.077	0.167	2592.386	−0.460	0.646	−0.403–0.250
Gender (*γ*_20_)	1.976	1.958	2588.152	1.009	0.313	−1.863–5.815
Chronic disease (*γ*_30_)	−0.917	2.030	2596.149	−0.451	0.652	−4.898–3.065
province-level variable						
Temperature (*γ*_01_)	−0.090	0.110	2592.392	−0.821	0.412	−0.306–0.125
Humidity (*γ*_02_)	0.042	0.137	2604.675	0.306	0.759	−0.227–0.312
Greening (*γ*_03_)	−0.071	0.247	2513.271	−0.288	0.773	−0.556–0.414
Control variable						
Traffic (*γ*_04_)	−1.156 × 10^−7^	7.219 × 10^−6^	24.342	−0.016	0.987	−1.50 × 10^−5^ –1.48 × 10^−5^
Provincial level moderating effects						
Age × Temperature (*γ*_11_)	0.000	0.002	2586.901	0.289	0.773	−0.003–0.004
Age × Humidity (*γ*_12_)	−0.000	0.002	2594.657	−0.172	0.863	−0.004–0.003
Age × Greening (*γ*_13_)	−0.000	0.004	2598.087	−0.123	0.902	−0.007–0.006
Gender × Temperature (*γ*_21_)	0.049	0.019	2581.089	2.661	0.008	0.013–0.086
Gender × Humidity (*γ*_22_)	−0.049	0.023	2589.550	−2.107	0.035	−0.093–−0.003
Gender × Greening (*γ*_23_)	0.018	0.041	2595.275	0.429	0.668	−0.063–0.099
Chronic disease × Temperature (*γ*_31_)	−0.001	0.020	2598.488	−0.038	0.970	−0.040–038
Chronic disease × Humidity (*γ*_32_)	−0.007	0.024	2603.483	−0.297	0.767	−0.055–0.040
Chronic disease × Greening (*γ*_33_)	0.036	0.044	2601.822	0.831	0.406	−0.049–0.122

## Data Availability

The data presented in this study are openly available in the following websites: (1) The China Family Panel Studies (http://isss.pku.edu.cn, accessed on 18 June 2021); (2) The National Meteorological Information Center of China (http://data.cma.cn, accessed on 15 February 2022); (3) The National Bureau of Statistics of China (http://www.stats.gov.cn, accessed on 18 June 2021).

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
