# Peer review of "The Associations of Meteorological and Environmental Factors with Memory Function of the Older Age in Urban Areas"

_ijerph, 2022, doi:10.3390/ijerph19095484_

Round 1

Reviewer 1 Report

This study investigated the associations of meteorological and environmental Factors with memory function of the older age in urban and evaluated the role of sex in this process. This study provided a theoretical basis for understanding the influence of meteorological environment factors on memory function of the elderly. It’s interesting as it addresses an important issue. I recommend this manuscript to be published with some revisions. Here is my comment:

Abstract: Well formulated.

Introduction:

  • The authors need to briefly introduce the mechanism and pathway of meteorological factors causing the decline of memory function in the elderly.
  • The authors need to optimize for Figure 1.
  • The authors need to state the novelty of this study and gather sufficient evidence to demonstrate its uniqueness from other studies.

Method:

  • Why are only the data of 2012 and 2016 in CFPS used and the data of 2013 to 2015 not used. Please explain further.
  • There are many kinds of meteorological factors, why not consider factors such as air pollution. Please explain further.

Results: Well formulated.

Discussion:

  • It should be noted that the influence of other factors on the memory function of the elderly, such as dietary structure, physical activity, mental state, and living environment, was not considered.

Reviewer 2 Report

  • The title of the article is grammatically incorrect. It should read "in urban areas" or "in urban regions"; "urban" is an adjective and not a noun.
  • The arguments that the authors have presented linking meteorological factors and cognitive performance in the elderly are unconvincing and do not take into account more plausible explanations, such as reduced air pollution in association with green spaces, or vitamin D deficiency in relation to increased time spent indoors. If they wish to make a more solid argument based on this, they could cite papers such as the one by Zhao et al. (2021) examining the link between ambient temperature and cognition in elderly women.
  • The authors have used the average temperature as an independent variable in their analysis; however, it is more likely that extremes of temperature are of relevance in reference to cognition.
  • A rationale for the inclusion of humidity as an independent predictor of cognitive performance should be provided, given that there is no empirical evidence linking the two.
  • Details of the memory test used in the CFPS, its items / domains covered, its validity / reliability, and its utility in identifying cognitive deficits in the target population should be provided. If these have been published in detail elsewhere (in a prior paper by the authors) then a reference to the same should be provided in the current paper.
  • The authors have used traffic as an independent variable. However, traffic may be a surrogate marker for various forms of air pollution, each of which has its own impact on human cognition. This should be acknowledged as a limitation and discussed in more depth when interpreting the results. (See Calderon-Garciduenas et al., 2022).
  • The authors have used linear analysis models throughout the study. It would be useful to explore the possibility of non-linear associations between environmental / climatic factors and cognition (e.g. using a curve fitting function in R or SPSS). This would add to the value of the paper.
  • The Discussion should attempt to explore the mechanistic links between meteorological variables and human cognition, as well as the impact of confounding factors that may not have been considered in this study.
  • References no. 35 and no. 53 should be provided in the standard format with regards to capitalization.

Reviewer 3 Report

I think this is an interesting question/research topic. I think it is well written.

My biggest problem with the paper is the scientific design. How can you possibly weed out all the factors that contribute to cognitive function? Yes, I believe that meteorological and environmental factors are big components, but there are many things that contribute to cognitive function. There is no causality examination or anything that tries to isolate factors. I am not sure this is possible. 

How do you plan to address this? 

Round 2

Reviewer 2 Report

The revised version of the manuscript is satisfactory in my opinion. No further major changes or corrections are required.

Author Response

Thank you for your helpful feedback and comments on our manuscript. We believe these help have greatly improved the manuscript and make it more suitable for publication in IJERPH.

Reviewer 3 Report

I understand the reply by the authors. However, if this is going to be their response then they need to do more in the paper to address it - not just a bit on shortcomings. I think this will need to be much more explicit and extended then what they have already done and indicated.
